# COMT and Neuregulin 1 Markers for Personalized Treatment of Schizophrenia Spectrum Disorders Treated with Risperidone Monotherapy

**DOI:** 10.3390/biom14070777

**Published:** 2024-06-29

**Authors:** Mariana Bondrescu, Liana Dehelean, Simona Sorina Farcas, Ion Papava, Vlad Nicoras, Dana Violeta Mager, Anca Eliza Grecescu, Petre Adrian Podaru, Nicoleta Ioana Andreescu

**Affiliations:** 1Department of Neurosciences-Psychiatry, “Victor Babes” University of Medicine and Pharmacy, Eftimie Murgu Square 2, 300041 Timisoara, Romania; mariana.bondrescu@umft.ro (M.B.); papava.ion@umft.ro (I.P.); 2Timis County Emergency Clinical Hospital “Pius Brinzeu”, Liviu Rebreanu 156, 300723 Timisoara, Romania; vnicoras@gmail.com (V.N.); dana.m3@yahoo.com (D.V.M.); ancagrecescu@gmail.com (A.E.G.); 3Doctoral School, “Victor Babes” University of Medicine and Pharmacy, Eftimie Murgu Square 2, 300041 Timisoara, Romania; 4Discipline of Medical Genetics, Department of Microscopic Morphology, Center of Genomic Medicine “Victor Babes” University of Medicine and Pharmacy, Eftimie Murgu Square 2, 300041 Timisoara, Romania; farcas.simona@umft.ro (S.S.F.); andreescu.nicoleta@umft.ro (N.I.A.); 5Faculty of Mathematics and Informatics, West University of Timisoara, Vasile Parvan 4, 300223 Timisoara, Romania; petre.podaru98@e-uvt.ro

**Keywords:** COMT, neuregulin 1, schizophrenia spectrum disorders, risperidone monotherapy

## Abstract

Pharmacogenetic markers are current targets for the personalized treatment of psychosis. Limited data exist on COMT and NRG1 polymorphisms in relation to risperidone treatment. This study focuses on the impact of COMT rs4680 and NRG1 (rs35753505, rs3924999) polymorphisms on risperidone treatment in schizophrenia spectrum disorders (SSDs). This study included 103 subjects with SSD treated with risperidone monotherapy. COMT rs4680, NRG1 rs35753505, and rs3924999 were analyzed by RT-PCR. Participants were evaluated via the Positive and Negative Syndrome Scale (PANSS) after six weeks. Socio-demographic and clinical characteristics were collected. COMT rs4680 genotypes significantly differed in PANSS N scores at admission: AG>AA genotypes (*p* = 0.03). After six weeks of risperidone, PANSS G improvement was AA>GG (*p* = 0.05). The PANSS total score was as follows: AA>AG (*p* = 0.04), AA>GG (*p* = 0.02). NRG1 rs35753504 genotypes significantly differed across educational levels, with CC>CT (*p* = 0.02), and regarding the number of episodes, TT>CC, CT>CC (*p* = 0.01). The PANSS total score after six weeks of treatment showed a better improvement for TT<CT genotypes (*p* = 0.01). NRG1 rs3924999 genotypes revealed GG<AG (*p* = 0.02) for PANSS G scores after six weeks, with AG and GG requiring higher doses (*p* = 0.007, *p* = 0.02). Overall, our study suggests that the genetic polymorphisms COMT rs4680, NRG1 rs35753505, and rs3924999 significantly impact the treatment response to risperidone in patients with SSD.

## 1. Introduction

Research on pharmacogenetic markers for personalized treatment in patients with psychosis, particularly those treated with risperidone, is a significant area of study in current psychiatry. Pharmacogenetics explores how genetic variants influence an individual’s response to drugs, including their efficacy and side effects [1]. Considering genetic studies, several genetic polymorphisms have been shown to play a role in the treatment response to antipsychotic medication in patients with schizophrenia [2,3].

Risperidone is a second-generation antipsychotic that has been extensively studied in treating psychosis such as schizophrenia. Previous data showed its superiority in treating both positive and negative symptoms compared to haloperidol [4,5]. Moreover, several studies have shown that risperidone can significantly impact relapse prevention [6,7] and treatment response in patients with early-onset schizophrenia, together with olanzapine and aripiprazole [8]. While effective in treating psychotic symptoms, risperidone treatment has been frequently linked to extrapyramidal side effects [9,10], tardive dyskinesia, weight gain [11], and elevated prolactin. Despite the fact that risperidone seems to increase prolactin levels [12,13], it has significantly fewer extrapyramidal side effects [4,14] than conventional antipsychotic treatment and shows positive effects on the negative symptoms of schizophrenia [15,16,17].

Psychoses from the schizophrenia spectrum (SSD) are extremely heterogenous and of great complexity, with important impacts on patients’ quality of life [18]. The prevalence is still reported, considering schizophrenia as a representative of the SSD class, to be around 1% [19]. The heterogeneous clinical presentation caused by complex disruptions in brain development originates in the interplay of genetic and environmental factors [20]. The core symptoms of schizophrenia spectrum disorders may be conceptualized in clusters, such as positive symptoms, negative symptoms and cognitive deficits, symptoms related to disorganized thought processes or behavior, and catatonic symptoms [21]. 

Negative symptoms of schizophrenia comprise heterogenous neuropsychological impairments [22], such as reduced thought content and spontaneity, blunted affect, and avolition, which greatly impact functionality, quality of life, and amount to important treatment challenges [23]. Several studies have shown a connection between negative symptoms of schizophrenia and a reduction in the prefrontal cortex, temporal cortex, right parietal cortex, limbic system, caudate nucleus, and corpus callosum volumes [24,25]. The reduction in dendritic arborization in the prefrontal cortex of patients with schizophrenia was linked to the appearance of negative symptoms related to psychosis [26,27]. 

Among genetic polymorphisms, neuregulin 1 (NRG1) was studied as a risk factor for schizophrenia [28,29,30,31,32]. Not only was it found to play a significant role in neuron development in the central nervous system [33,34] but also to be a pathophysiological marker of schizophrenia [35], particularly due to its complex involvement in neuron migration, myelinization, trophic effect on neurons and glial cells, and modulation of both glutamatergic and GABA-ergic synaptic transmission [36]. NRG1 modulatory effects are transduced by ErbB2, ErbB3, and ErbB4 receptors. Data suggest that NRG1-erbB4 is a signaling pathway of particular importance in schizophrenia [37,38]. Thus, NRG1 through the ErbB4 receptor plays a crucial role in the complex processes of learning and memory [39], impacting the synaptic plasticity in schizophrenia [40], which accounts for cognitive deficits, emotion dysregulation, and impaired global functioning. Interestingly, risperidone showed important effects in improving negative symptoms of schizophrenia [41,42] and decreasing cortical atrophy in murine models [43]. A study by Yang et al. found that positive improvements in attention and reasoning in patients suffering from schizophrenia, after 12 weeks of risperidone treatment, were associated with the rs3924999 and rs35753505 polymorphisms of NRG1 [42]. 

Another important susceptibility marker of schizophrenia with a significant impact on treatment response and cognitive performance seems to be the gene encoding catechol-O-methyltransferase (COMT) enzyme located on chromosome 22q12 [44]. COMT regulates dopamine availability in the prefrontal cortex [45]. Among many COMT polymorphisms, rs4680 and rs4818 were frequently associated with schizophrenia. While rs4818 polymorphisms seem to affect COMT expression, rs4680 polymorphisms seem to be involved in enzyme activity. Moreover, COMT Val158Met (rs4680) was related to treatment response and treatment resistance in patients suffering from schizophrenia [46]. In the COMT Val158Met polymorphism, the amino acid valine (Val) is replaced with the amino acid methionine (Met) at codon 158 of membrane-bound and at codon 108 of the soluble short form of COMT. Consequently, a decrease in enzyme activity of three- to fourfold was observed in A (Met) carriers with favorable response to antipsychotics [47], while the G (Val) variants increase enzyme activity, resulting in lower dopamine levels in the prefrontal cortex and lower response [48]. Thus, COMT Val158Met rs4680 genotypes influence enzyme activity, MetMet (AA) genotypes exhibit low enzymatic activity, MetVal (AG) genotypes show intermediate and more stable enzymatic activity, while ValVal (GG) genotypes exhibit the highest enzymatic activity, with all types impacting dopamine levels dominantly in the prefrontal cortex [49,50]. Regarding clinical implications, treatment response in schizophrenia spectrum disorder patients showed significant differences related to COMT Val158Met polymorphisms [51]. COMT rs4680 was particularly associated with risperidone efficacy, showing general improvements in symptoms in patients diagnosed with schizophrenia [52]. 

Although pharmacogenetic markers related to schizophrenia and clinical outcomes of antipsychotic treatment are areas of intense study, relatively few data are available on risperidone monotherapy in relation to both COMT and NRG1 polymorphisms. Therefore, the main outcome of the present study is to evaluate the treatment response to risperidone measured by PANSS in relation to COMT (rs4680) and NRG1 (rs3924999 and rs35753505) polymorphisms.

## 2. Materials and Methods

### 2.1. Study Design

Sample size was calculated in accordance with the reported prevalence of schizophrenia in the general population of 1% [19], with a 99% confidence level and a 5% margin of error, which resulted in a minimum of 27 participants to reach significance for the study’s population. Statistical power was established at 80%.

An observational study was performed with subjects diagnosed with SSD, according to DSM-5 criteria, treated with risperidone monotherapy. One hundred and seven voluntarily admitted patients were recruited from the Psychiatric Ward of “Pius Brinzeu” County Emergency Hospital in Timisoara, Romania, from November 2021 to January 2024. The inpatients enrolled in the study were between the ages of 19 and 74 years old, admitted and treated for at least six weeks with risperidone monotherapy. Inpatients treated with other antipsychotic medication that was afterward initiated on risperidone monotherapy and enrolled in the study underwent a period of a 7-day washout of the previous medication. Thus, the T0 was considered accordingly. The decision to receive risperidone as an antipsychotic was established by specialized psychiatrists from the hospital, independent of this study. Only then, patients were asked and informed about the study, and if accepted to take part and sign the informed consent, were enrolled. The eligibility criteria for the study were as follows: (1) patients meeting criteria for psychosis from the schizophrenia spectrum in an acute episode of psychosis or with chronic psychosis being diagnosed for at least three years, (2) receiving risperidone as an antipsychotic for at least six weeks with proper washout periods, (3) agreed to take part in the study and signed an informed consent form. The risperidone treatment was orally administered, and the dosage adjustments were realized blinded from the genotypes by L.D. There was a gradual increase in dose from a low dose to a therapeutic dose according to patients’ needs during the time of the hospitalization. The dosages were stabilized within one week (2-8 mg/day) according to standard clinical protocols. Exclusion criteria included the following: (1) patients undergoing other antipsychotic treatment or with unclear washout period from previous antipsychotic medication (less than seven days), (2) subjects treated with other medication that could interfere with the accurate evaluation of the genotype’s impact on the treatment, (3) subjects with comorbid diseases interfering with the treatment outcomes (neurological disorders, renal and hepatic disorders), (4) subjects with drug-induced psychosis or those with a history of drug use one month prior to admission. Following the first stage of evaluation, one patient was excluded due to incomplete protocol, two patients dropped out of the study because they needed to switch to another antipsychotic, and one was excluded by reason of unclear diagnosis. Thus, only 103 subjects underwent the data analysis and proceeded through the study. Socio-demographic and clinical data were gathered from the subjects. 

PICO model for the current study consisted of the following: P (population): SSD patients according to DSM-5 diagnostic guideline; I (intervention): risperidone monotherapy; C (comparison): patients with AA>AG>GG for rs4680 (COMT), patients with TT>TC>CC for rs35753505, and patients with AA>AG>GG for NRG1 rs3924999 (NRG1); and O (outcomes): treatment response (reported using PANSS) according to genetic polymorphisms.

### 2.2. Ethics

Subjects included in the study were informed about the purpose of the research and its implications and gave their written informed consent. The present study was performed according to the Helsinki Declaration Guidelines for scientific experiments involving human subjects. Ethical approval number 10/2021 was obtained from the Ethical Board of the “Victor Babes” University of Medicine and Pharmacy Timisoara.

### 2.3. Clinical Assessment

Patients’ evaluations by PANSS were performed on admission and after six weeks of risperidone monotherapy in a blinded fashion to the genotyping by two researchers (M.B and L.D). The evaluation scale comprises 30 items. Each item is evaluated on a scale of 7 degrees, from 1 (absent) to 7 (extreme). The questionnaire has three subscales, summing up to a total score (PANSS T): Positive Scale (PANSS P) (7 items), Negative Scale (PANSS N) (7 items), and General Psychopathology Scale (PANSS G) (16 items). The ranging scores are between 30 and 280 points for the total score, between 7 and 49 points for each Positive and Negative subscale, and between 16 and 112 points for the General subscale. The internal consistency for PANSS was adequate, with Cronbach’s alpha’s between 0.70 and 0.85 [53,54].

Participants were grouped into two categories based on the reduction in PANSS scores calculated as the difference between total scores at admission and total scores after six weeks of risperidone treatment, as follows: non-responders with a reduction less than 25% in PANSS total score and responders with a reduction in PANSS total score of 25% or more. The threshold for the PANSS total score reduction was established based on existing data on treatment response in patients with SSD [44,55]. Thus, the threshold for treatment response was established at a 25% or more reduction in PANSS total score. 

### 2.4. COMT and NRG1 Genotyping

A 2 mL volume of peripheral blood was drawn for pharmacogenetic analysis on admission. The isolation of the genomic DNA was conducted in accordance with the manufacturer’s procedures of the MagCore Nucleic Acid Extraction Kit (RBC Bioscience, New Taipei City, Taiwan). DNA was stored at -20 degrees Celsius. Epoch Microplate Spectrophotometer (Agilent BioTek EPOCHSN) was used to determine the DNA concentration. For the current study, one variant (rs4680, Assay ID: C__25746809_50) of the COMT gene and two variants (rs35753505, Assay ID: C____216209_50 and rs3924999, Assay ID: C____359159_10) of NRG1 genes were selected for the analysis. We used TaqMan Drug Metabolism Genotyping Assays (Applied Biosystems) and TaqMan Genotyping Master Mix (Applied Biosystems) according to manufacturer protocol. Purified DNA was amplified in a real-time polymerase chain reaction (PCR) on the LightCycler 480 (Roche; Basel, Switzerland). Gene Scanning software version 1.5.1 (Roche) was used. The genotyping process was conducted in a double-blind manner by two laboratory personnel. For the quality control of the genotyping, 5% of the samples were chosen randomly. The genotyping of ten samples was repeated, with a reproducibility of 100%. Patients were grouped according to the specific genotype for each single nucleotide polymorphism (SNP), as follows: for COMT rs4680, the genotypes were AA, AG, and GG; for NRG1 rs3924999, the genotypes were AA, AG, and GG; and for NRG1 rs35753505, genotypes were CC, CT, and TT. 

### 2.5. Statistic Analysis

Descriptive and inferential statistics were performed using programming language R, version 4.3.3, Environment Software: R studio 2023.12.0+369 “Ocean Storm” Release (33206f75bd14d07d84753f965eaa24756eda97b7, 17.122.2023) for Windows; Mozilla/5.0, Chrome/116.0.5845.190 Electron/26.2.4 Safari/537.36. Continuous data were represented as mean and standard deviation, while categorical variables were represented as absolute and percentage values. Shapiro–Wilk test was used to assess the distribution of the variables. For non-Gaussian distribution data, the Kruskal–Wallis test was performed to determine the differences between groups, while for normally distributed variables, the ANOVA test was performed. For groups that showed significant differences, post hoc Dunn Chi-squared test was performed for data that followed non-Gaussian distribution and the Tukey HSD test for data with Gaussian distribution to identify pairs of groups with significant differences. Statistical resampling techniques were used to perform bootstrapping and permutation tests for a more robust estimation of a small sample size, followed by the specific test mentioned above in accordance with the distribution of the data. These methods are in line with further robust genetic research, allowing for reliable inference despite the imbalanced data [56,57]. Outlier tests were performed to identify data points that significantly deviate from the rest of the data. The Levene test was also performed to assess the equality of variances for variables calculated for two or more groups. Statistical power was chosen for 95% and a significance threshold at α = 0.05. 

## 3. Results

### 3.1. Descriptive Analysis

In this study, 103 subjects treated with risperidone monotherapy were included. Genetic polymorphisms of rs4680 (AA; AG; GG), rs35753505 (CC; CT; TT), and rs3924999 (AA; AG; GG) were evaluated and are represented in Figure 1. 

Participants were divided into two categories based on the reduction in PANSS total scores after six weeks of risperidone treatment, as follows: non-responders (the reduction in PANSS total score was less than 25%) and responders (the reduction in PANSS total score was 25% or more). 

Socio-demographical data for the two groups are presented in Table 1. The mean age was 40 years old for both non-responders (SD = 13) and responders (SD = 14) (*p* = 0.95). Considering sex, 51% of non-responders were male and 49% were female, similar to the responder group, where 52% were male and 48% were female (*p* = 1). The mean for years of education was identical for responders and non-responders, consisting of 12 years of studies (SD = 3) (*p* = 0.73).

Regarding occupation, there was no significant difference between the groups (p=0.06); 6% of non-responders were students compared to 9% of responders, and 34% of non-responders in comparison to 44% of responders were employed. In the unemployed category, 34% were non-responders and 20% were responders. Considering retirement status, 2% of non-responders and 7% of responders were retired, while 34% of non-responders and 20% of responders were retired due to ill health. No statistical differences in the socio-demographical parameters were found between the categories.

The clinical characteristics of the population are presented in Table 2. The mean age of onset of the disease was 33 years old (SD = 13), being the same for both responders and non-responders. The mean number of psychotic episodes was 4 (SD = 5) for non-responders and 3 (SD = 3) for responders (*p* = 0.49). Considering the dose of risperidone per day, the mean dose for non-responders was 4 (SD = 1) and for responders was 5 (SD = 2) (*p* = 0.32). 

With respect to diagnostics, 49% of non-responders and 52% of responders were in the first episode of psychosis, receiving a diagnosis of schizophreniform disorder. No statistically significant difference was found between diagnostic categories in relation to treatment response (*p* = 0.49).

When comparing mean scores of PANSS between non-responders and responders, while no significant differences were present between the means of non-responders and responders on admission for all the scores (P T0, *p* = 0.09), (N T0, *p* = 0.54), (G T0, *p* = 0.10), (T T0, *p* = 0.23), significant results were found after 6 weeks of risperidone treatment for all the scores, except for those of the N subscale. 

Regarding genetic polymorphisms, no significant difference was found in relation to the treatment response category for any of them (COMT rs4680, *p* = 0.21, NRG1 rs35753505, *p* = 0.55, NRG1 rs3924999, *p* = 0.77). Interestingly, the most frequent genotype was the same for non-responders and responders, being represented by AG for COMT rs4680, TT for NRG1 rs35753505, and AG for NRG1 rs3924999.

### 3.2. Impact of Genetic Polymorphisms on Clinical Characteristics

Statistical analysis on the impact of the combined action of COMT rs4680/NRG1 rs3924999 genotypes on the age of onset showed significant differences between the genotypes (H = 12.34, *p* = 0.03). After Dunn’s post hoc test, the patients exhibiting the AA (COMrs4680)/AA (NRG1 rs3924999) genotypes showed higher mean ages than the patients exhibiting the AG (COMrs4680)/AG (NRG1 rs3924999) genotypes (*p*-adj = 0.009) Similarly, the patients exhibiting AA (COMrs4680)/AA (NRG1 rs3924999) showed higher mean ages than the patients exhibiting GG (COMrs4680)/AG (NRG1 rs3924999) (*p*-adj = 0.027), suggesting that patients with the combined genotypes AA for COMT rs4680 and AA for NRG1 rs3924999 may have a later onset of psychosis than those with either AG/AG genotypes or AG/GG genotypes of COMT rs4680/NRG1 rs3924999.

The Kruskal–Wallis test showed differences close to significance between NRG1 rs3934999 genotypes and the dose of risperidone per day (Chi-squared = 5.9, df = 2, *p* = 0.05), suggesting further investigation should be considered. Similarly, the Kruskal–Wallis test showed significant differences between NRG1 rs35753505 and the level of education (*p* = 0.04), with no significance after the post hoc Dunn test (*p*-adj > 0.05). 

Following the grouping of the genotypes, unequal groups were obtained for COMT rs4680 (30 AA, 56 AG, 17 GG), for NRG1 rs35753505 (7 CC, 38 CT, 58 TT), and for NRG1 rs3924999 (15 AA, 51 AG, 37 GG). To address the challenges posed by these unequal group sizes and to ensure robust statistical analysis, bootstrapping and permutation methods were employed with 1000 iterations. Consequently, for COMT rs4680, iterations were made with 17 GG and 17 randomly for each AG and AA over 1000 times; for NRG1 rs35753505, iterations were made with seven CC and seven random iterations for each CT and TT; and for NRG1 rs3924999, iterations were made with 15 AA and 15 random for each AG, and GG. The results showed significance, with a dominance of *p* < 0.05, which is represented in Figure 2.

The bootstrapping and permutation model when analyzing the level of education in relation to NRG1 rs35753507 showed significant differences between the genotypes revealed in Figure 3. Statistical analysis of the genotypes of NRG1 rs35753507 showed significant differences in relation to patients’ levels of education, doses of risperidone, and number of episodes, and the results are presented in Figure 4. The Kruskal–Wallis Chi-squared test showed significant differences between NRG1 rs35753505 genotypes and the level of education, with CC<CT after the post hoc Dunn test (*p*-adj = 0.02); thus, subjects with CT genotypes may have better levels of education.

Following post hoc Dunn, significant differences were found between NRG1 rs35753505 genotypes and the dose of risperidone per day, with TT>CC (*p*-adj = 0.04), suggesting that patients with TT genotypes may need higher doses of treatment. Likewise, significant differences were found for NRG1 rs3924999 genotypes, with AG>AA (*p*-adj = 0.007) and GG>AA (*p*-adj = 0.02), suggesting that AG and GG genotypes may require higher dosages of risperidone.

Statistical analysis on the number of episodes and NRG1 rs35753505 genotypes showed significant differences after the post hoc Dunn test, suggesting that CT>CC (*p*-adj = 0.01) and TT>CC (*p*-adj = 0.01). Consequently, patients with TT and CT genotypes may experience more frequent episodes of psychosis than those with CC genotypes.

### 3.3. Impact of Genetic Polymorphisms on Treatment Response

Statistical analysis of PANSS N scores on admission (PANSS N T0) and COMT rs4680 polymorphisms showed significant differences between AA and AG genotypes (Dunn post hoc Z-value = −2.52, *p*-adj = 0.03). Consequently, patients with AG genotypes had significantly higher scores of PANSS N on admission than patients with AA genotypes. 

Following ANOVA analysis of ΔPANSS G (PANSS G T0-PANSS G T6) after risperidone treatment and COMT rs4680 genotypes, significant variability was found between AA, AG, and GG (df = 2, F-value = 3.002, *p* = 0.053). Further, Tukey’s HSD test was performed to identify which pair of genotypes had statistically significant differences, and the results are presented in Table 3.

Although no significant differences were found between AG and AA (*p*-adj = 0.2) or between GG and AG (*p* = 0.46) on the improvement in PANSS G subscale sores, a borderline significant difference was found for GG and AA genotypes of COMT rs4680 with AA>GG (*p*-adj = 0.05), suggesting that AA genotypes may have better improvements on the PANSS G subscale than GG genotypes after risperidone treatment.

Following the Kruskal–Wallis test, significant differences were found between genotypes of COMT rs4680 and ΔPANSS N (PANSS N T0-PANSS N T6) after risperidone treatment (Chi-squared = 6.68, dg = 2, *p* = 0.03). A Dunn post hoc test pairwise comparison was used to determine which specific groups of COMT rs4680 genotypes differ from each other, and the results are presented in Table 4.

Significant differences were found between AA and AG genotypes of COMT rs4680 and ΔPANSS N, suggesting that patients with AG could have better improvements on PANSS N scores after risperidone treatment, AG>AA (Z-value = −2.52, *p*-adj = 0.03). 

Statistical analysis of the PANSS total score after 6 weeks of treatment (PANSS T T6) showed significant differences between genotypes of COMT rs4680 (*p* = 0.01), and the results are presented in Figure 3. 

Following post hoc Dunn statistical analysis, significant differences were found with AG > AA (*p*-adj = 0.04) and GG >AA (*p*-adj = 0.02), suggesting that subjects with AG and GG genotypes may have lower improvements in the PANSS total score after 6 weeks of treatment with risperidone than those with the AA genotype. Similarly, significant differences were found for NRG1 rs35753505 genotypes in relation to PANSS T T6 (*p* = 0.01). Following post hoc Dunn statistical analysis, significant differences were found with TT>CT (*p*-adj = 0.01).

Statistical analysis showed significant differences between the genotypes of COMT rs4680, NRG1 rs3924999, and NRG1 rs35753505 and PANSS subscale scores after six weeks of risperidone treatment and are represented in Figure 5.

Regarding PANSS P subscale scores after six weeks of treatment (PANSS P T6), significant differences were found for COMT rs4680 genotypes following the post hoc Dunn test, with AG>AA subgroups (*p*-adj = 0.04), where AG genotypes seem to have a lower reduction in positive symptoms than AA genotypes after risperidone. The results are presented in Figure 6.

Following statistical analysis of PANSS G subscale scores after six weeks of treatment (PANSS G T6) and NRG1 rs35753505 and NRG1 rs3924999, significant differences following the post hoc Dunn showed that TT>CT (*p*-adj = 0.02) NRG1 rs35753505 and GG>AG (*p*-adj = 0.02) of NRG1 rs3924999, underlining that TT and GG genotypes may experience lower reductions in the general symptomatology after risperidone.

## 4. Discussion

The current study examined the relation between the genetic polymorphisms of COMT rs4680, NRG1 rs35753505, NRG1 rs3924999, and treatment response to risperidone in patients with psychosis from the schizophrenia spectrum. Treatment adherence during the six weeks of risperidone monotherapy was assessed by psychiatrists by direct observation during hospital admission. Moreover, the clinical characteristics of the patients were also studied considering the genotypes. To our knowledge, this is the first study on the Romanian population, examining both COMT and NRG1 genotypes regarding treatment response to risperidone as antipsychotic monotherapy in patients with schizophrenia spectrum disorders.

Two other studies have found a connection between COMT rs4680 and improvements in the PANSS total score in patients treated with risperidone [46,52]. Specifically, patients with AA genotypes (homozygotes for methionine) (Met) showed the best response after eight weeks of risperidone in the PANSS total score and PANSS N score, followed by the heterozygotes Val158 Met (AG genotype) [46,58]. In line with these, the current study suggested that patients with AG genotypes may respond better to risperidone than those with AA genotypes, as they showed a significant improvement in PANSS N scores after six weeks of treatment. On the other hand, in our study, subjects with AG genotypes had significantly higher scores of PANSS N on admission than patients with AA genotypes, which is supported by a previous study with similar results [59]. Moreover, our data suggested that patients with AG or GG genotypes of COMT rs4680 showed a lower reduction in PANSS total scores after six weeks of risperidone than those with AA genotypes (homozygotes for Methionine), which is in line with previous studies. However, this is not the first study to find connections between low-activity genotypes (AA) and the severity of positive symptoms in schizophrenia [60,61]; some other studies also related GG genotypes with more severe negative symptoms [62], while others showed better response on PANSS N for AG and AA genotypes than for GG genotypes after risperidone treatment [52]. Similar results to ours were revealed in a study by Perkovic et al., where Met carriers treated with olanzapine showed significantly better improvements in PANSS total scores [47]. 

A less studied perspective is that of the impact of COMT rs4680 genotypes on PANSS G scores after risperidone monotherapy. In our study, the differences in PANSS G scores from admission to six weeks of treatment showed a borderline significant difference between GG and AA genotypes of COMT rs4680 (*p*-adj = 0.05), suggesting that AA genotypes may have better improvements on the PANSS G subscale than GG genotypes after risperidone treatment. Regarding PANSS P scores after six weeks of treatment, significant differences were found for COMT rs4680 genotypes, where AG genotypes seem to have a lower reduction in positive symptoms than AA genotypes after risperidone. Our results are similar to other studies, which observed more severe positive symptoms and poor response to antipsychotic medication in patients with GG genotypes [63]. An interesting study by Chhabra et al., evaluating both COMT and NRG1 rs35753505 genotypes, showed better improvements in the positive symptoms for patient homozygotes for Val, especially when associated with AA genotypes of NRG1 after transcranial direct current stimulation [64]. 

Although the role of neuregulin 1 in schizophrenia is strongly related to complex alterations with an impact on plasticity and its regulating role in excitatory/inhibitory synapses, especially in the prefrontal cortex [65], little is known about its role in treatment response. In our study, NRG1 rs35753505 genotypes were significantly correlated with the PANSS total score after six weeks of risperidone monotherapy. Thus, patients with CT genotypes showed better improvements in the PANSS total score than patients with TT genotypes. On the other hand, patients with TT genotypes and GG genotypes of NRG1 rs3924999 showed significantly lower reductions in PANSS G scores after six weeks of risperidone treatment than those with CT and GG genotypes, respectively. Similar results correlated TT genotypes of NRG1 with non-responder status in patients treated with first-generation antipsychotics [66]. Another study showed a connection between C carriers of NRG1 rs35753505 and a decreased expression of the NR2C subunit in the right cerebellum as a compensatory mechanism for the hypofunction of the NMDA receptor of patients suffering from schizophrenia [67]. This finding explains the cognitive deficits observed in schizophrenia by the hypofunction of glutamatergic synapses in the prefrontal cortex, thalamus, and cerebellum. A study by Moradkhani et al. found a correlation between CC-genotypes of NRG1 rs35753505 and PANSS total score and subscale scores, the same genotype showing a significant association with cognitive dysfunctions in patients with schizophrenia [68]. This could be explained by the fact that subjects with C alleles seem to have lower NRG1 levels in comparison with those with TT genotypes, as suggested by a study on preterm infants [69]. Notwithstanding the fact that several data found an association between genotypes of NRG1 and deficits in cognition [42], we could not establish any influence of NRG1 genotypes on the PANSS N subscale, which could be an indicator of cognitive and social impairments.

Considering the importance of NRG1 in neuronal growth and synaptic plasticity, interesting associations of several polymorphisms of NRG1 were observed in the current study. Thus, subjects with CT genotypes of rs35753505 completed significantly more years of studies than those with CC genotypes. On the contrary, patients with TT and CT genotypes seem to present more frequent psychotic decompensation. A study by Yang et al. showed that antipsychotic medication could increase Neuregulin 1 serum levels in patients suffering from schizophrenia, thus improving psychotic symptoms [70]. What is more, the number of psychotic episodes significantly impacts symptom severity and the likelihood of remission in those patients. Consequently, the greater the number of psychotic episodes, the more significant the brain loss, resulting in exacerbating psychotic symptoms and reducing the chances for remission periods over time [71]. However, only those with TT genotypes from this study needed significantly higher doses of treatment. Likewise, patients with AG and GG genotypes of NRG1 rs392499 also required higher dosages of risperidone.

An interesting result of our research was that patients with the combined genotypes of AA for COMT rs4680 and AA for NRG1 rs3924999 may have a later onset of psychosis than those with either AG/AG or AG/GG of COMT rs4680/NRG1 rs3924999. This could suggest the important interaction of COMT and NRG1 in the pathogenesis of schizophrenia spectrum disorders.

Despite several contradictions, these results should be considered in the context of the complex pathogenesis of schizophrenia, where, apart from the intricate genetic mechanisms involved [72,73,74], there may be interplay between several other factors, such as anatomic factors, biological factors, and early life events, thus altering treatment response. As underlined before, environmental factors may interact with genetic variability, resulting in alterations in the clinical manifestation, and the treatment results in psychosis from the schizophrenia spectrum. 

Several limitations should be considered in the current study. Firstly, the sample size was relatively small, especially after grouping patients according to the genotypes. Secondly, patients were evaluated only once, after six weeks of risperidone treatment, while other studies performed repeated evaluations. Thirdly, therapeutic drug monitoring was not evaluated in the current study. Although versatile, with an important bias reduction and precise, bootstrapping and permutation tests should be interpreted with caution in the context of small sample sizes.

## 5. Conclusions

This study explored the relationship between the genetic polymorphisms COMT rs4680, NRG1 rs35753505, and NRG1 rs3924999 and the response to risperidone treatment in Romanian patients with schizophrenia spectrum disorders. The findings indicated that patients with AG genotypes of COMT rs4680 showed significant improvements in negative symptoms (PANSS N scores) but had higher initial symptom severity compared to the AA genotypes. Furthermore, AA genotypes had a better overall symptom reduction (PANSS total scores). Additionally, AG genotypes showed a lower reduction in positive symptoms (PANSS P scores) than AA genotypes. NRG1 rs35753505 CT genotypes were associated with better total score improvements, while TT genotypes of NRG1 rs3924999 were linked to a lower reduction in general symptoms (PANSS G scores). This study highlighted the complex interplay of genetic factors in treatment response and suggested that combined COMT and NRG1 genotypes might influence the onset of psychosis. Despite some contradictory findings, the results underscore the intricate genetic and environmental factors impacting schizophrenia treatment outcomes.

## Figures and Tables

**Figure 1 biomolecules-14-00777-f001:**
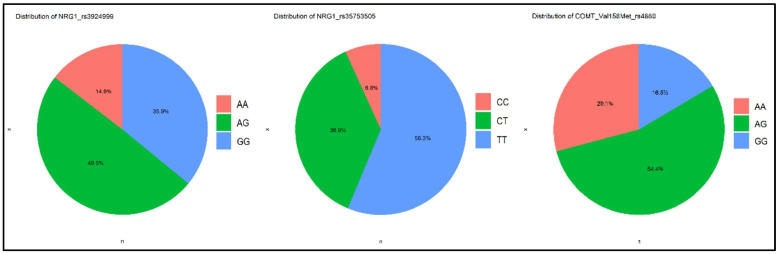
Genetic polymorphisms of the group. Patients with psychosis from schizophrenia spectrum and genetic polymorphisms groups of COMT rs4680 (AA; AG; GG), of NRG1 rs35753505 (CC; CT; TT) and of NRG1 rs3924999 (AA; AG; GG).

**Figure 2 biomolecules-14-00777-f002:**
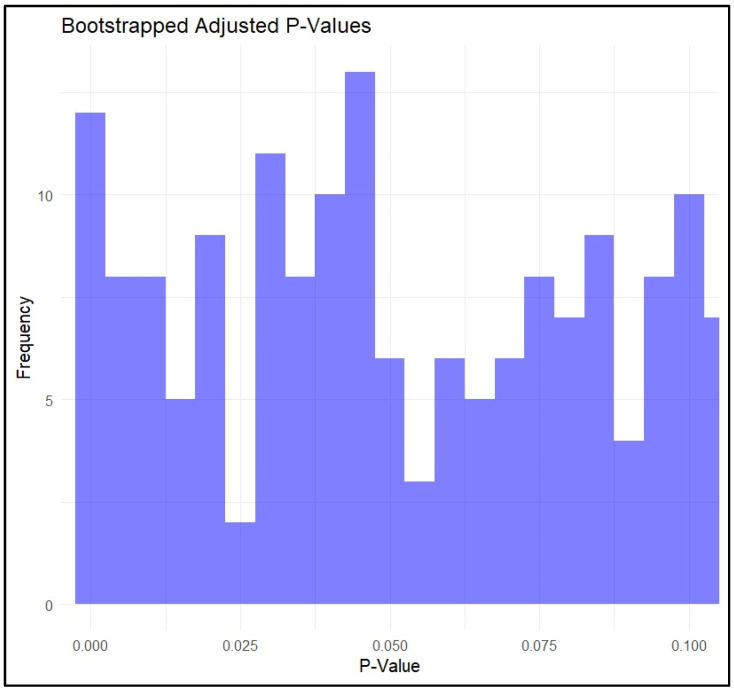
Bootstrapping adjusted *p*-values for the genotypes.

**Figure 3 biomolecules-14-00777-f003:**
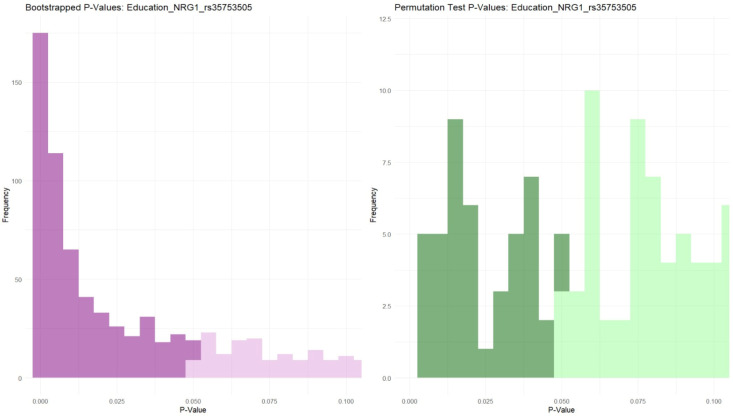
NRG1 rs35753505 genotypes related to the level of education. The different nuances of the colors displayed are in correspondence with the degree of statistical significance (i.e., the darker shades symbolize a statistically significant value, *p* < 0.05).

**Figure 4 biomolecules-14-00777-f004:**
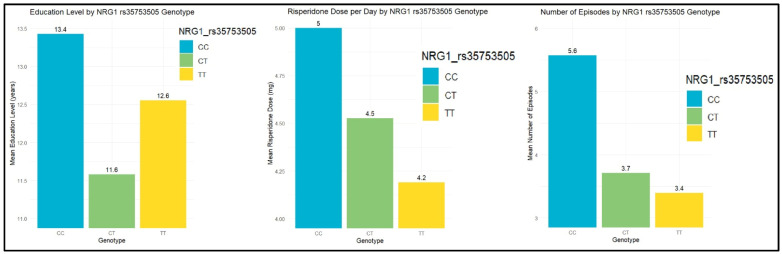
Genotypes of NRG1 and clinical characteristics. Patients with psychosis from schizophrenia spectrum and genetic polymorphisms groups of COMT rs4680 (AA; AG; GG), of NRG1 rs35753505 (CC; CT; TT) and of NRG1 rs3924999 (AA; AG; GG).

**Figure 5 biomolecules-14-00777-f005:**
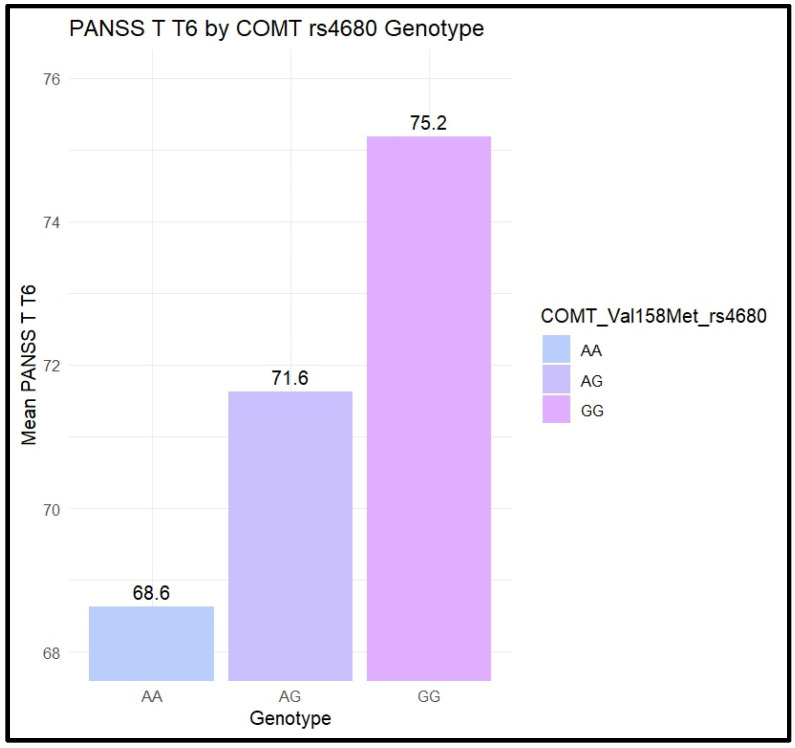
PANSS total score after six weeks of risperidone treatment in relation with COMT rs4680 genotypes. COMT rs4680 (AA; AG; GG), PANSS T T6= PANSS total score after six weeks of treatment.

**Figure 6 biomolecules-14-00777-f006:**
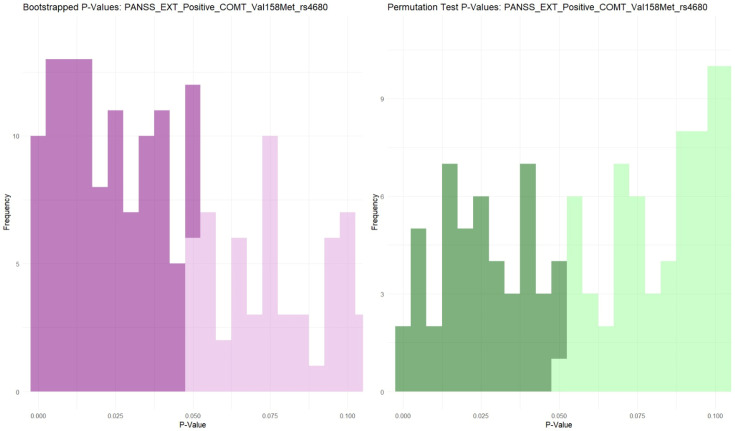
PANSS P T_6_ in relation to COMT rs4680. The different nuances of the colors displayed are in correspondence with the degree of statistical significance (i.e., the darker shades symbolize a statistically significant value, *p* < 0.05).

**Table 1 biomolecules-14-00777-t001:** Socio-demographics of the population.

Variables	Non-Responders (n = 47)	Responders (n = 56)	*p*-Value
**Age** (mean ± SD)	40 ± 13	40 ± 14	0.95
**Sex**	47	56	1
Male	51% (24)	52% (29)	
Female	49% (23)	48% (27)	
**Education (years of study)** (mean ± SD)	12 ± 3	12 ± 3	0.73
**Occupation**			0.06
Student	6% (3)	9% (5)	
Employed	34% (11)	44% (25)	
Unemployed	34% (16)	20% (11)	
Retired	2% (1)	7% (4)	
Ill-health Retired	34% (16)	20% (11)	

Treatment response was based on the PANSS threshold described in the materials and methods; T_0_- PANSS total score and subscales scores on admission, T_6−_PANSS total score and subscales scores after six weeks of risperidone treatment; SD—standard deviation; α value after Bonferroni correction = 0.012.

**Table 2 biomolecules-14-00777-t002:** Clinical characteristics of the population.

Variables	Non-Responders (n = 47)	Responders (n = 56)	*p*-Value
**Age of onset (years)** (mean ± SD)	33 ± 13	33 ± 13	0.77
**Number of episodes** (mean ± SD)	4 ± 5	3 ± 3	0.49
**Dose/day (mg)** (mean ± SD)	4 ± 1	5 ± 2	0.32
**Diagnostic**			0.49
Schizophreniform disorder	49% (23)	52% (29)	
Schizophrenia	17% (8)	21% (12)	
Schizoaffective disorder	15% (7)	18% (10)	
Delusional Disorder	19% (9)	9% (5)	
**PANSS T_0_** (mean ± SD)			
PANSS P _T0_	26 ± 6	24 ± 6	0.09
PANSS N _T0_	19 ± 6	21 ± 7	0.54
PANSS G _T0_	48 ± 9	51 ± 9	0.10
PANSS T _T0_	94 ± 14	97 ± 15	0.23
**PANSS _T6_** (mean ± SD)			
PANSS P _T6_	21 ± 5	15 ± 3	0.001
PANSS N _T6_	17 ± 5	15 ± 5	0.05
PANSS G _T6_	41 ± 8	35 ± 8	0.001
PANSS T _T6_	80 ± 13	64 ± 10	0.001
**COMT (rs4680)**	47	56	0.21
AA	12 (26%)	18 (32%)	
AG	24 (51%)	32 (57%)	
GG	11 (23%)	6 (11%)	
**NRG1 (rs35753505)**			0.55
CC	3 (6%)	4 (7%)	
CT	20 (43%)	18 (32%)	
TT	24 (51%)	34 (61%)	
**NRG1 (rs3924999)**			0.77
AA	6 (13%)	9 (16%)	
AG	25 (53%)	26 (46%)	
GG	16 (34%)	21 (38%)	

Treatment response was based on the PANSS threshold described in the materials and methods; T_0_- PANSS total score and subscales scores on admission, T_6−_PANSS total score and subscales scores after six weeks of risperidone treatment; SD—standard deviation; α value after Bonferroni correction = 0.01.

**Table 3 biomolecules-14-00777-t003:** Tukey’s HSD test for COMT rs4680 genotypes and the improvement in PANSS G subscale scores after risperidone.

*ΔPANSS G for Genotypes of COMT rs4680	diff	lwr	upr	*p*-adj
AG-AA	−3.05	−7.29	1.18	0.2
GG-AA	−5.65	−11.34	0.03	0.05
GG-AG	−2.59	−7.78	2.58	0.46

*ΔPANSS G (PANSS G T0-PANSS G T6) after risperidone; Diff = difference; lwr = lower bound; upr = upper bound; *p*-adj = adjusted *p*-value.

**Table 4 biomolecules-14-00777-t004:** Dunn’s post hoc test for COMT rs4680 genotypes and the improvement in PANSS N subscale scores after risperidone.

*ΔPANSS N for Genotypes of COMT rs4680	Z-Value	*p*-adj
AA-AG	−2.52	0.03
AA-GG	−1	0.89
AG-GG	0.92	1

*ΔPANSS N (PANSS N T0-PANSS N T6) after risperidone; *p*-adj = adjusted p value.

## Data Availability

Data available on request.

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
