# Peer review of "COMT and Neuregulin 1 Markers for Personalized Treatment of Schizophrenia Spectrum Disorders Treated with Risperidone Monotherapy"

_biomolecules, 2024, doi:10.3390/biom14070777_

Round 1
Reviewer 1 Report
Comments and Suggestions for Authors
Bondrescu et al. present the results of an observational study probing the possible association of COMT and NRG1 polymorphisms and treatment outcomes with a 6-week risperidone course among Schizophrenia Spectrum Disorders (SSD) patients experiencing symptoms exacerbation and requiring inpatient care. Patients were recruited from a single center from 2021 to 2024. The results and the soundness of the applied methodology appear solid. The present manuscript is certainly current and of interest, especially in light of the necessity to test Pharmacogenomic markers (PGX) on varied populations and in different settings. More specifically, in SSD, there is a dearth of evidence regarding their potential worth. Therefore, the timing for this report appears particularly appropriate.
My concerns are minor and are summarized in the following bullet points.
It is my understanding that recruited subjects have been admitted and treated for 6 weeks as inpatients. However, this is only an inference on my part, as I do not think I have seen it literally specified. Authors may consider specifying this element.
-If indeed all the patients remained in the ward for the 6-week treatment course, we might consider the adherence to treatment relatively optimal (but not necessarily). Consider adding in the discussion what measures researchers adopted to assess treatment adherence rate while on the ward (e.g., direct observation?) or outside if indeed part of the treatment course was taken as an outpatient. However, even in the best circumstances, interindividual differences may still be at play in influencing risperidone blood levels with the same oral doses. I assume therapeutic drug monitoring for risperidone was not available, and therefore, it cannot be added to the discussion. However, I believe it might be worthwhile adding it as a possible limitation (e.g., could not account for pharmacokinetic/adherence differences).
-It is unclear what might be the impact of disease duration and number of previous psychotic episodes or the number of past failed treatments in the presented sample. This element was apparently not explored. If relevant data for this element is available, please consider adding it to the results. As far as I am concerned, even a brief passage detailing the comparison of responsive vs unresponsive patients might suffice to this end.
-Considering the requirement for providing informed consent to participate, I assume that at the time of recruitment, all patients were voluntarily admitted to the psychiatric ward. If I am wrong in assuming this, please consider specifying the number of subjects sectioned at the beginning of the inpatient stay that have been subsequently accepted hospital admission or, indeed, the number of patients involuntarily admitted. Moreover, I was wondering whether there might be any difference in the overall hospital admission duration or the timing of risperidone treatment initiation during the hospitalisation itself. Exclusionary criteria included the presence of current antipsychotic treatment other than risperidone, but maybe a switch to risperidone would have been appropriate even during the same inpatient stay (e.g., unclear washout period from other antipsychotic eventually practised).
-Figure 1 and Figure 3 labels for the three panels included in each figure appear too small. I wonder whether authors may consider switching the orientation of the grid vertically rather than horizontally and increasing the font size. This way, it would be possible to improve the overall clarity of the figures.
Author Response
|
Response to Reviewer 1 Comments
|
||
|
1. Summary |
|
|
|
Thank you very much for taking the time to review this manuscript. Please find the detailed responses below and the corresponding revisions in track changes in the re-submitted manuscript. |
||
|
3. Point-by-point response to Comments and Suggestions for Authors Bondrescu et al. present the results of an observational study probing the possible association of COMT and NRG1 polymorphisms and treatment outcomes with a 6-week risperidone course among Schizophrenia Spectrum Disorders (SSD) patients experiencing symptoms exacerbation and requiring inpatient care. Patients were recruited from a single center from 2021 to 2024. The results and the soundness of the applied methodology appear solid. The present manuscript is certainly current and of interest, especially in light of the necessity to test Pharmacogenomic markers (PGX) on varied populations and in different settings. More specifically, in SSD, there is a dearth of evidence regarding their potential worth. Therefore, the timing for this report appears particularly appropriate. Comments 1: It is my understanding that recruited subjects have been admitted and treated for 6 weeks as inpatients. However, this is only an inference on my part, as I do not think I have seen it literally specified. Authors may consider specifying this element. Response 1: Thank you for your thoughts and observations. We added some clarification in the Materials and Methods section, Study design on this matter: “The inpatients enrolled in the study were between the age of 19 and 74 years old, admitted and treated for at least 6 weeks with risperidone monotherapy. Inpatients treated with other antipsychotic medication, that were afterwards initiated on risperidone monotherapy and enrolled in the study underwent a period of a 7-day washout of the previous medication, thus the T0 was considered accordingly. The decision of receiving risperidone as antipsychotic was established by specialized psychiatrists from the hospital, independent from the study. Only then, patients were asked and informed about the study and if accepted to take part and sigh the informed consent, were enrolled.” (lines 126-133 currently) Comments 2: -If indeed all the patients remained in the ward for the 6-week treatment course, we might consider the adherence to treatment relatively optimal (but not necessarily). Consider adding in the discussion what measures researchers adopted to assess treatment adherence rate while on the ward (e.g., direct observation?) or outside if indeed part of the treatment course was taken as an outpatient. However, even in the best circumstances, interindividual differences may still be at play in influencing risperidone blood levels with the same oral doses. I assume therapeutic drug monitoring for risperidone was not available, and therefore, it cannot be added to the discussion. However, I believe it might be worthwhile adding it as a possible limitation (e.g., could not account for pharmacokinetic/adherence differences).
Response 2: Thank you for your suggestion. We agree with you, but due to financial constraints, we were unable to determine the concentration of the drug. Our study is self-funded, and the costs would exceed 100 euro per patient. Therefore, we added this as limitation as well, and clarified the method of assessing drug adherence in discussion section accordingly: “Thirdly, therapeutic drug monitoring was not evaluated in the current study.” (lines 496-497). Comments 3: -It is unclear what might be the impact of disease duration and number of previous psychotic episodes or the number of past failed treatments in the presented sample. This element was apparently not explored. If relevant data for this element is available, please consider adding it to the results. As far as I am concerned, even a brief passage detailing the comparison of responsive vs unresponsive patients might suffice to this end. Response 3: Thank you for your observation. We clarify the difference between responsive and unresponsive patients (lines 185-187 currently): “Thus, the threshold for treatment response was established at 25% or more reduction in PANSS total score”, and we added literature data to explain the relevance of our findings on this matter. (lines 473-479 currently):” A study by Yang et al showed that antipsychotic medication could increase Neuregulin 1 serum levels in patients suffering from schizophrenia, thus improving psychotic symptoms (70). What is more, the number of psychotic episodes significantly impact symptoms severity and the likelihood of remission in those patients. Consequently, the greater the number of psychotic episodes, the most significant brain loss resulting in exacerbating psychotic symptoms and reducing the chances for remission periods in time (71).” |
||
|
Comments 4: -Considering the requirement for providing informed consent to participate, I assume that at the time of recruitment, all patients were voluntarily admitted to the psychiatric ward. If I am wrong in assuming this, please consider specifying the number of subjects sectioned at the beginning of the inpatient stay that have been subsequently accepted hospital admission or, indeed, the number of patients involuntarily admitted. Moreover, I was wondering whether there might be any difference in the overall hospital admission duration or the timing of risperidone treatment initiation during the hospitalisation itself. Exclusionary criteria included the presence of current antipsychotic treatment other than risperidone, but maybe a switch to risperidone would have been appropriate even during the same inpatient stay (e.g., unclear washout period from other antipsychotic eventually practised).
|
||
|
Response 4: Thank you for pointing this out. We understand your concerns. In the hospital where the study took place all patients are voluntarily admitted. Patients admitted had different hospital durations, we assess them from T0 (first day of risperidone monotherapy or after 7 days washout period) and after 6 weeks of risperidone, although some of them required longer hospitalization. We added some clarification in the Materials and methods section accordingly: “107 voluntarily admitted patients were recruited from the Psychiatric Ward of “Pius Brinzeu” County Emergency Hospital from Timisoara, Romania, from November 2021 to January 2024. The inpatients enrolled in the study were between the age of 19 and 74 years old, admitted and treated for at least 6 weeks with risperidone monotherapy. Inpatients treated with other antipsychotic medication, that were afterwards initiated on risperidone monotherapy and enrolled in the study underwent a period of a 7-day washout of the previous medication, thus the T0 was considered accordingly. The decision of receiving risperidone as antipsychotic was established by specialized psychiatrists from the hospital, independent from the study. Only then, patients were asked and informed about the study and if accepted to take part and sigh the informed consent, were enrolled. The eligibility criteria for the study were: (1) patients meeting criteria for a psychosis from the schizophrenia spectrum in acute episode of psychosis or with chronic psychosis being diagnosed for at least 3 years, (2) receiving risperidone as antipsychotic for at least 6 weeks with proper washout periods, (3) agreed to take part in the study and signed an informed consent. The risperidone treatment was orally administered, and the dosages adjustments were realized blinded from the genotypes by L.D. There was a gradual increase of dose from a low dose to a therapeutic dose according to patients needs during the time of the hospitalization. The dosages were stabilized within 1 week (2-8 mg/day) according to standard clinical protocols. Exclusion criteria included: (1) patients undergoing other antipsychotic treatment or with unclear washout period from previous antipsychotic medication (less than 7 days)” (lines 123-146)
|
||
|
Comments 5: -Figure 1 and Figure 3 labels for the three panels included in each figure appear too small. I wonder whether authors may consider switching the orientation of the grid vertically rather than horizontally and increasing the font size. This way, it would be possible to improve the overall clarity of the figures.
|
||
|
Response 5: Thank you for your observation. We adjusted the label to improve the clarity.
|
||

Reviewer 2 Report
Comments and Suggestions for Authors
The authors investigated the impact of the polymorphisms COMT rs4680 and NRG1 rs35753505 and rs3924999 on risperidone monotherapy of schizophrenia patients. Participants of the study were evaluated by PANSS after 6 weeks. A main finding was that PANSS total score showed an improvement for TT>CC genotypes. The obtained results confirm the outcome if earlier studies by others.
Concerns:
1. Line 92: Please rephrase the sentence: While..., while....
2. Several other studies... You show only two references (46,52). So, you better write two other studies.
3. Line 284. Kruskal-Wallis test was showed... Please correct (only showed)
Comments on the Quality of English Languagesee above.
Author Response
|
Response to Reviewer 2 Comments
|
||
|
1. Summary |
|
|
|
Thank you very much for taking the time to review this manuscript. Please find the detailed responses below and the corresponding revisions in track changes in the re-submitted manuscript. |
||
|
3. Point-by-point response to Comments and Suggestions for Authors The authors investigated the impact of the polymorphisms COMT rs4680 and NRG1 rs35753505 and rs3924999 on risperidone monotherapy of schizophrenia patients. Participants of the study were evaluated by PANSS after 6 weeks. A main finding was that PANSS total score showed an improvement for TT>CC genotypes. The obtained results confirm the outcome if earlier studies by others. Concerns: Comments 1: Line 92: Please rephrase the sentence: While..., while.... Response 1: Thank you for your useful observation. We corrected accordingly. Comments 2: Several other studies... You show only two references (46,52). So, you better write two other studies. Response 2: Thank you for your suggestion. We adjusted. Comments 3: Line 284. Kruskal-Wallis test was showed... Please correct (only showed) Response 3: Thank you for your remark. We corrected. |
||
|
|
||
